# Autophagy in Hematological Malignancies

**DOI:** 10.3390/cancers14205072

**Published:** 2022-10-17

**Authors:** Olga García Ruiz, José Manuel Sánchez-Maldonado, Miguel Ángel López-Nevot, Paloma García, Angelica Macauda, Francisca Hernández-Mohedo, Pedro Antonio González-Sierra, Manuel Martínez-Bueno, Eva Pérez, Fernando Jesús Reyes-Zurita, Daniele Campa, Federico Canzian, Manuel Jurado, Juan José Rodríguez-Sevilla, Juan Sainz

**Affiliations:** 1Genomic Oncology Area, GENYO, Centre for Genomics and Oncological Research: Pfizer/University of Granada/Andalusian Regional Government, PTS Granada, 18016 Granada, Spain; 2Instituto de Investigación Biosanitaria de Granada (ibs.GRANADA), Complejo Hospitales Universitarios de Granada, Universidad de Granada, 18016 Granada, Spain or; 3Department of Immunology, University of Granada, 18016 Granada, Spain; 4Campus de la Salud Hospital, PTS Granada, 18007 Granada, Spain; 5Genomic Epidemiology Group, German Cancer Research Center (DKFZ), 69120 Heidelberg, Germany; 6Hematology Department, Virgen de las Nieves University Hospital, 18014 Granada, Spain; 7Genomic Medicine Area, GENYO, Centre for Genomics and Oncological Research: Pfizer/University of Granada/Andalusian Regional Government, PTS Granada, 18016 Granada, Spain; 8Department of Biochemistry and Molecular Biology I, University of Granada, 18071 Granada, Spain; 9Department of Biology, University of Pisa, 56126 Pisa, Italy; 10Department of Medicine, University of Granada, 18071 Granada, Spain; 11Department of Hematology, Hospital del Mar, 08003 Barcelona, Spain

**Keywords:** autophagy, hematological malignancies, hematopoiesis, therapeutic target, autophagy-related variants, clinical outcomes, disease progression, patient survival

## Abstract

**Simple Summary:**

Autophagy is a dynamic and tightly regulated process that seems to have dual effects in cancer. In some contexts, it can induce carcinogenesis and promote cancer cell survival, whereas in others, it acts preventing tumor cell growth and tumor progression. Thus, autophagy functions seem to strictly depend on cancer ontogenesis, progression, and type. Here, we will dive into the current knowledge of autophagy in hematological malignancies and will highlight the main genetic components involved in each cancer type.

**Abstract:**

Autophagy is a highly conserved metabolic pathway via which unwanted intracellular materials, such as unfolded proteins or damaged organelles, are digested. It is activated in response to conditions of oxidative stress or starvation, and is essential for the maintenance of cellular homeostasis and other vital functions, such as differentiation, cell death, and the cell cycle. Therefore, autophagy plays an important role in the initiation and progression of tumors, including hematological malignancies, where damaged autophagy during hematopoiesis can cause malignant transformation and increase cell proliferation. Over the last decade, the importance of autophagy in response to standard pharmacological treatment of hematological tumors has been observed, revealing completely opposite roles depending on the tumor type and stage. Thus, autophagy can promote tumor survival by attenuating the cellular damage caused by drugs and/or stabilizing oncogenic proteins, but can also have an antitumoral effect due to autophagic cell death. Therefore, autophagy-based strategies must depend on the context to create specific and safe combination therapies that could contribute to improved clinical outcomes. In this review, we describe the process of autophagy and its role on hematopoiesis, and we highlight recent research investigating its role as a potential therapeutic target in hematological malignancies. The findings suggest that genetic variants within autophagy-related genes modulate the risk of developing hemopathies, as well as patient survival.

## 1. Introduction

Hematological malignancies are a group of blood diseases originating from a clonal expansion of hematopoietic cells [1]. Hematopoiesis is a highly orchestrated process whereby a pool of hematopoietic stem cells (HSCs) and self-renewing precursor cells initiate the lymphoid and myeloid lineages [2]. With age, HSC pools become poorer in terms of absolute cell numbers and cell diversity, leading to deregulated proliferation and differentiation and, thus, tumor development [3]. Historically, hematological neoplasms have been treated with conventional chemotherapy drugs and, more recently, with combined therapies. However, despite the broad number of therapeutic options available, many treatments eventually generate partial or total resistance in many patients, leading to drug insensitivity in cancer cells and, consequently, treatment ineffectiveness [4]. In an attempt to elucidate the cellular and molecular mechanisms underlying the process of resistance to therapy, numerous investigations have identified potential therapeutic targets in multiple cellular functions [5]. 

Among these biological processes, autophagy has attracted increasing attention as it constitutes an adaptive cell survival process whereby the cell, under stress conditions such as lack of nutrients or oxidative stress, begins an essential degradative pathway involving lysosomes [6] that help to maintain cell homeostasis [7,8]. There are two types of autophagy: micro-autophagy, which consists of the recycling of damaged or badly processed molecules, and macro-autophagy (herein referred to as autophagy), which recycles damaged or dysfunctional organelles [9]. This review is focused on macro-autophagy, which consists of several successive stages. 

Numerous studies have described the classic role of autophagy in the maintenance of homeostasis and cellular metabolism, as an alternative survival process to apoptosis or senescence [10]. In exchange for sacrificing organelles or damaged molecules, energy is obtained, and the cell cytoplasm is preserved, thus achieving a cytoprotective effect that, however, can become a mechanism of cell death if exacerbated [8]. Recently, autophagy has also been shown to regulate other important cell processes, such as cellular differentiation, cellular death, and the cell cycle, playing a key role in tumorigenesis and chemoresistance in hematological malignancies [5,11]. Autophagy plays an important role in hematopoietic differentiation, which constitutes a widely studied process, likely because blood cells are easy to purify and have a well-defined progenitor tree. Autophagy is involved in cell remodeling during terminal differentiation, long-term maintenance of cell type, and regulation of the balance between self-renewal and quiescence in stem cells [12]. Thus, it can act as an antitumoral mechanism thanks to the control of damaged organelles, free oxygen radicals (ROS), antigen presentation by dendritic cells, and inflammation, avoiding cytotoxicity and DNA damage [13]. However, it may also have a protumor role, since, once the tumor is initiated, autophagy is capable of reducing the cellular damage caused by drugs, making cancer cells insensitive to treatment and thus promoting resistance to standard chemotherapies [14]. In fact, its dysregulation can cause genetic damage and promote tumor development through an uncontrolled proliferative phenotype in the different cells involved in hematopoiesis [14,15]. Even once the tumor is formed, dysregulated autophagy can serve as an adaptive mechanism to stabilize oncogenic proteins, which would otherwise be degraded by correct autophagic machinery. In hematological neoplasms, the role of autophagy seems to depend on the context, varying not only according to the type of tumor but also the type of progenitor and the specific phase in which it acts (initiation vs. progression), and it may have opposite roles [16]. All this makes autophagy a very interesting therapeutic target in hematological cancers [9,14].

In this review, we describe the most recent findings in the study of autophagy in hematological malignancies and discuss its role in hematopoiesis and the crosstalk between autophagy and the proteasome. We also review the mechanisms of autophagic regulation that have been investigated in recent years and that constitute an alternative and/or improvement to the current chemotherapeutic treatments. Lastly, we summarize the main findings regarding the impact of autophagy-related variants on the risk of developing hematological malignancies, as well as their effect on modulating patient survival and clinical complications. 

## 2. Autophagic Machinery and Process

The autophagy mechanism begins due to different forms of cellular stress, such as nutrient deprivation of growth factors, hypoxic conditions, the presence of reactive oxygen species (ROS) or protein aggregates, damage to DNA or organelles, and even contact with intracellular pathogens [17]. Signaling pathways dependent on specific stimuli or more general molecular mechanisms are activated to regulate the different phases of autophagy, which normally acts in coordination with other cellular control processes, such as apoptosis or the proteasome machinery [17,18]. The process of autophagy takes place in a series of well-established steps. First, a portion of the intracellular membrane is isolated, giving rise to a phagophore. Next, nucleation and elongation occur, in which the phagophore closes by fusing its ends to create a double-membrane vesicle called an autophagosome, which traps the content to be degraded. In the subsequent maturation step, the autophagosome fuses with endosomes to become an amphisome, which later fuses with lysosomes. Once this occurs, an autophagolysosome is formed, capable of enzymatically metabolizing intravesicular content that is degraded and discharged into the cytoplasm to be reused by the cell to obtain energy and promote the initiation of different biological processes. 

### 2.1. Initiation

One of the most important pathways acting in the regulation of autophagy involves the mTOR complex, a serine/threonine kinase that forms two functionally distinct complexes: mTORC1, which is dependent on rapamycin and regulates cell size, and mTORC2, independent of rapamycin, which modulates the actin of the cytoskeleton and, therefore, the shape of the cell [5]. mTORC1 plays a fundamental role in the response to nutrient deprivation [19], especially a lack of amino acids and growth factors [20]. In conditions of the abundance of nutrients, mTORC1 is found in the lysosome, activated by its Rheb subunit, inhibiting autophagy. Rapamycin, an inhibitor of mTORC1, therefore, behaves as a strong inducer of autophagy, even under conditions rich in nutrients [21]. mTORC1 interacts with the ULK1 complex, consisting of ULK1, ATG13, ATG101, and FIP200 (Figure 1). All of these subunits are crucial for the initiation of autophagy and the behavior of the complex. In normal conditions, mTORC1 phosphorylates ULK1 and ATG13, inactivating the complex and inhibiting autophagy. However, under conditions of nutrient deprivation, mTORC1 is inhibited and dissociates from the ULK1 complex, causing activation of ULK1. This dissociation occurs in response to the action of the AMPK kinase, which inhibits mTORC1 by phosphorylating two of its subunits, Rheb and RAPTOR. AMPK then phosphorylates ULK1 and activates it, after which ULK1 phosphorylates ATG13 and FIP200, activating them as well. Complete activation of the complex leads to phagophore formation or membrane isolation. Lastly, ULK1 phosphorylates Beclin-1 within the PI3K complex, which facilitates the initiation of the autophagy pathway [5,14,20] (Figure 1).

### 2.2. Nucleation and Elongation

Nucleation of the autophagosome is triggered by the formation of a class III PI3K complex consisting of VPS34, Beclin-1 (ATG6), ATG14L, and p150 (VPS15) [22]. The formation of this complex is regulated by Beclin-1 (located in the membrane of the endoplasmic reticulum), which binds to UVRAG or to different members of the Bcl-2 and BclXL family to activate or inhibit autophagy, respectively. The activity of this class III complex is modulated by ULK1, through phosphorylation of Beclin-1 and AMBRA1, a protein that binds to Beclin-1 during the process [23]. Once it is formed, the complex PI3K produces PIP3 (3-phosphatidyl inositol phosphate), which operates as an initial signal for the phagophore formation, recruiting PIP3-binding proteins such as WIPI that, in turn, are capable of recruiting other ATG proteins to the autophagosomal membrane that come from the endoplasmic reticulum, the Golgi apparatus, and the mitochondria [24]. Once ATG9 (the only transmembrane protein, located mainly in the trans membrane of the Golgi apparatus and in endosomes) is recruited to the phagophore, it contributes to the transport of lipids to the site of autophagosome formation. Lastly, ATG14L, as part of the PI3K complex, is responsible for initiating the recruitment of other ATG proteins involved in the elongation of the phagophore and for promoting its transformation into an autophagosome. ATG14L facilitates the recruitment of the two ubiquitin-like conjugation systems, ATG12–ATG5–ATG16L and LC3-II (ATG8), which, in turn, facilitate the expansion of the double-membrane structure and the binding of membrane ends to form the autophagosome (Figure 1). LC3-II remains associated with the autophagosomal membrane until fusion with the lysosome occurs. When this occurs, LC3-II proteins that were on the outer membrane dissociate, while those that were on the inner membrane are degraded with the remainder of the intravesicular content [2,5,14,20,25]. Although autophagy generally exhibits an indiscriminate degradation system, there is evidence showing that it possesses some substrate selectivity. LC3-II interacts with an adapter molecule, SQSTM1/P62, which contains ubiquitin-binding domains, thus trapping ubiquitinated proteins and binding them to LC3-II [26]. This type of autophagy is called LC3-associated phagocytosis, but there are also other highly selective ones such as those mediated by chaperones [2,12,27].

### 2.3. Autophagolysosome Formation

Once the autophagosome is formed, it binds to Rab7 endosomes to form the amphisome, which fuses to lysosomes to form the autophagolysosome. UVRAG is involved in this step, activating the Rab7 GTPase protein, which regulates the fusion of autophagosome and lysosome through LC3/GABARAP [28]. In order to mature, other proteins, such as LAMP1/2, must be activated in addition to Rab7. This step can be inhibited by Rubicon, which binds to Beclin-1, p150, PI3k, and UVRAG proteins. Microtubules also act to facilitate the fusion process. Finally, the content is degraded thanks to the hydrolytic activity, and the decomposition products are released to cytosol by permeases [14,20,25]. 

### 2.4. Autophagy Regulatory Drugs

There are multiple regulatory elements that act in different steps of the process, activating or inhibiting autophagy and, thus, affecting the development of different hematologic malignancies, among other diseases [14,15,29,30,31]. The molecular mechanisms of action of the autophagy drugs vary according to the disease type, tumor stage, the drug used, and even the step of autophagy affected by the drug, as they act in coordination with other essential cell survival processes, such as apoptosis. However, in general, there are two main types of action of the autophagy-related therapies; some of them promote the initiation of autophagy and activate autophagic cell death, whereas other drugs inhibit it, restoring chemosensitivity in cells with drug resistance [32].

Regarding autophagy-inducing molecules, many of them are anticancer agents. The first to be identified was rapamycin, an inhibitor of the mTOR complex, which acts in a similar way to conditions of nutrient deficiency. A derivative of rapamycin, everolimus, works in the same way. Another molecule that works by inhibiting mTOR is Torin1, which activates autophagy much more than rapamycin [33]. On the other hand, one of the best-known inducers is metformin, which acts by activating the AMPK kinase, not only inhibiting mTOR but also activating ULK1 [20]. Arsenic trioxide and tetrandin, a medicinal herbal extract, can also activate autophagy, leading to cell death mediated by this process [34]. Fascaplisin, an alkaloid derived from a marine sponge, achieves the same effect [35].

Regarding inhibitors, the most relevant drugs are chloroquine (CQ) and hydroxychloroquine (HCQ), preventing lysosomal acidification in an oral dosage and possessing an adequate safety profile [36]. Other molecules, such as 3-methyladenine (3-MA) or wortmannin, inhibit autophagic flow, although, due to their side-effects in acting on other cellular pathways and their low solubility, they are not usually considered the most suitable option [36]. Bafilomycin A1 also compromises autophagy by inhibiting the proton transport in the lysosome and by inhibiting hydrolases [36]. Lastly, there are other nonpharmacological mechanisms that act to prevent autophagy. Some of these alternatives are the use of siRNA to silence the expression of autophagy genes or the use of histone deacetylases inhibitors (HDACs) [14]. Many of these therapies are being studied in preclinical trials in combination with conventional chemotherapy or targeted therapy [36,37,38].

## 3. The Crosstalk between Autophagy and Proteasome

The ubiquitin–proteasome system (UPS) is the cellular mechanism responsible for the degradation of damaged proteins, ensuring cellular homeostasis. It has long been considered an independent process, parallel to autophagy, sharing the same function. However, it is currently known that autophagy and the proteasome are connected, such that the regulation of one process influences the other [39,40,41]. This relationship constitutes an important aspect when treating some tumors, such as multiple myeloma [40] or mantle cell lymphoma [42].

UPS plays an important role in the control of cell growth, DNA replication and repair, metabolism, and host immune responses [43,44,45]. Damage to this machinery can trigger a variety of diseases, including the formation of tumors [46]. UPS selectively degrades cellular proteins, initially tagging them with ubiquitin, a homopolymeric protein of 76 amino acids that binds to various lysine residues. Ubiquitination can produce degradation signals, either via the proteasome system or via the autophagy machinery [39]. In the first case, the proteasome is the specific organelle that is responsible for carrying out this process. UPS consists of ubiquitinated molecules, ubiquitin-activating enzymes (E1), ubiquitin-conjugating enzymes (E2), and ubiquitin ligases (E3), as well as deubiquitinases and the proteasome [18]. It is located in both the cytosol and the nucleus, and it is directly responsible for the degradation of more than 80% of cellular proteins [41].

There are multiple ways in which proteasome and autophagy communicate with each other. First and most obvious is that both processes share substrates and regulators. Ubiquitin is one of the most important molecules in the two systems. In UPS, the three enzyme complexes work together to conjugate polyubiquitin chains to the proteins to be degraded. In selective autophagy, the p62 protein recognizes polyubiquitinated chains and recruits them to LC3-II proteins in the autophagosome [47]. Furthermore, mutations causing C-terminal extension in proteins usually degraded by the proteasome provoke an alternative degradation mediated by the autophagic pathway, since the high level of the structure makes impossible their entry through the proteasomal pores [48]. This also occurs with massively accumulated proteins, which are either degraded by macro-autophagy or survive temporarily as aggregates that are later targeted by lysosomes [41,48]. Additionally, when the proteasomal machinery is overwhelmed, a compensatory autophagy mechanism is activated that directs the accumulation of misfolded proteins to large aggregates, such that they are no longer specifically recognized by the UPS. In this interaction, the p62 protein and HDAC6 histone deacetylase play a very important role, since they are responsible for the formation of the aggregates and their allocation to the corresponding pathway [26]. Other molecules regulating the balance between proteasomal and autophagic degradation are the BAG family of proteins. The BAG3/BAG1 ratio determines what type of degradation occurs, since BAG1 transports misfolded proteins that are identified by chaperones to the proteasome, while BAG3 promotes the autophagic machinery [49]. However, the strongest evidence to date for the connection between autophagy and UPS comes from the fact that inhibition of UPS causes activation of autophagy as a compensation mechanism. It is known that, when there are too many incorrect proteins and proteasome activity is reduced, cells activate autophagy to protect the cell from the cytotoxic effects of the accumulation of these proteins [40]. This activation is mediated by the ATF4 transcription factor, which positively regulates autophagy genes [50]. In a parallel pathway, Bcl2 is separated from Beclin-1 thanks to phosphorylation by the IRE1–JNK147 pathway. It is also known that, in the event of UPS inhibition, p53 regulates autophagic activation through AMPK kinase, inhibiting the mTOR complex. Another way in which autophagy compensates for UPS inhibition is through stress on the endoplasmic reticulum (ER), which occurs when the UPS reduces its activity, which acts as an activator of autophagy [17,40]. On the contrary, there is no evidence of proteasome activation to compensate for autophagy inhibition. This happens because UPS cannot act as a protein breakdown mechanism when autophagy is blocked, since autophagosomal substrates are too large to overcome the proteasome barrier. In fact, an inhibited autophagy compromises the elimination of p62, resulting in its accumulation and consequent delay in the “sequestration” of ubiquitinated proteins sent to the proteasome [40].

The application of proteasome inhibitors (PIs) for the treatment of hematological malignancies has spread since the discovery of bortezomib (BTZ), used for the treatment of multiple myeloma (MM) [51,52,53] and mantle cell lymphoma (MCL) [54,55]. The promising results obtained with this drug in both diseases have prompted its use in the treatment of other hematological malignancies, such as acute leukemia. However, its broad use has been hampered by the development of drug resistance, as happens for many other PIs [18]. This resistance is thought to be produced, among other things, by the activation of compensatory autophagy pathways, which alleviate the cytotoxic effect of the drug and prevent cell death [18,51]. In fact, administering rapamycin in murine models protects cells from the cell death caused by proteasome inhibitors [56]. Consequently, the combination of proteasome and autophagy inhibitors seems to be the best treatment strategy to overcome therapy resistance in these pathologies, as discussed later.

## 4. The Role of Autophagy in Hematopoiesis

Hematopoiesis is a highly controlled physiological process via which all blood cells are produced and renewed from a small population of HSCs [57]. HSCs are found within the bone marrow in adult individuals, in a nutritive hypoxic niche that maintains its properties of quiescence, self-renewal, and differentiation. They are normally found in a quiescent state (G0 within the cell cycle); however, under specific stimuli, they can be activated, dividing symmetrically or asymmetrically, such that they are self-renewing or differentiating, respectively. Through this mechanism, HSCs are capable of producing differentiated progeny, giving rise to the lymphoid (T and B lymphocytes) and myeloid (erythrocytes, megakaryocytes, and granulocytes) lines [58].

The important position of the HSCs at the beginning of the progenitor tree highlights the significance of the cytoprotective pathways that act against damage accumulation and cellular dysfunctions such as mutations that can result in clonal hematopoiesis or hematological diseases. Autophagy is considered one of these cytoprotective pathways since it avoids stress generation and maintains hematopoietic homeostasis. In fact, several studies suggest that autophagy regulates the balance between quiescence, self-renewal, and differentiation of HSCs, in both normal and stress conditions [59,60,61]. Cheung et al. compared gene expression microarray data from numerous stem-cell types and identified a gene signature for quiescent hematopoietic stem cells (HSCs) that included the upregulation of several autophagic pathway genes, such as *ULK2*, *PINK1*, *ATG8* homolog *Gabarap11*, and *FOXO3A* [62]. *FOXO3A* is a transcription factor that maintains the quiescence of HSCs, regulates autophagy, and protects cells from proapoptotic stimulation through reprogramming of gene expression [63]. *FOXO3* is also known to induce the expression of autophagy genes, including *ATG4*, *LC3B*, *ULK2*, *VPS34*, *Beclin1*, *Map11c3b*, and *Bnip3* [63,64], and trigger autophagy through ROS, which activate FOXO factors causing high levels of sestrin, a protein that enhances the inhibition of mTORC1 mediated by the AMPK kinase pathway [65,66]. 

It is also known that a functional autophagic machinery is essential for the maintenance of the self-renewal/differentiation balance of HSCs. For example, it has been observed in mice that deletion of the *ATG7* gene causes severe damage to the self-renewal capacity of HSCs, since it prevents the formation of secondary colonies in both in vitro and in vivo experiments [67]. It has been observed that *ATG7*^−/−^ HSCs accumulate aberrant mitochondria, leading to increased ROS levels and increased DNA damage. In fact, ATG7^−/−^ mice showed severe anemia and a reduction in the total count of T, B, and NK lymphocytes (natural killer cells) in peripheral blood. Furthermore, *ATG7*^−/−^ mice have fewer HSCs, whereas they show an increase in myeloid progenitors, which ends up directing differentiation toward myelopoiesis [63]. Myeloid progenitors, unlike HSCs, react to *ATG7* deficiency by initiating an alternative autophagic pathway that compensates for the deletion and allows cellular survival [67], which seems to indicate a greater dependence of HSCs on autophagy.

A similar phenotype to the deletion of *ATG7* in HSCs is obtained by inducing the deletion of the *FIP200* gene [68]. This deletion is lethal in embryos and is also characterized by severe anemia. *FIP200* is required for the maintenance and function of fetal HSCs; its deletion does not cause an increase in apoptosis in the HSCs, but rather a greater proliferation that, in addition, is associated with a greater mitochondrial mass and higher levels of ROS [69]. These data suggest that HSCs are more vulnerable to autophagy failure than differentiated cells, which have a greater ability to survive. In fact, it is known that HSCs have high autophagy levels, which decrease as the degree of differentiation increases [63,70]. These levels have been measured in murine and human HSCs by specific autophagosome staining and colocalization of the LC3-II protein using a LAMP1 inhibitor, bafilomycin A [70].

Maintaining basal autophagy levels appears to be essential for HSCs, so that they can eliminate damaged mitochondria and ROS accumulation. The most important mechanism for this is mitophagy, responsible for maintaining quality control of mitochondria and preventing excess ROS from causing hyperproliferation of HSCs followed by their depletion due to apoptosis. Indeed, healthy HSCs have a much lower number of mitochondria than more differentiated cells, possibly due to the hypoxic environment in which HSCs are found, which favors glycolytic metabolism to achieve energy requirements, instead of the oxidative phosphorylation carried out by more differentiated cells [71,72]. As described, the hypoxic stem-cell niche could induce autophagy through the expression of Bnip3 dependent on the HIF1α factor [73]. In fact, inhibition of mTOR induces autophagy and also contributes to cell quiescence by repressing mitochondrial activity and promoting glycolysis [74]. This is essential to maintain stem-cell properties, since, during differentiation, there is a change in cellular metabolism from anaerobic glycolysis toward mitochondrial oxidative phosphorylation, and altering this step ahead of time can change the decision to engage in self-renewal or differentiation fate from HSCs [75].

In general, the existence of defects in autophagy damages HSCs (initially through an increase in oxidative stress that later causes alterations in DNA and produces genomic instability), thus promoting the appearance of hematological tumors. It is, therefore, necessary to explore new strategies to restore the proper functioning of autophagy in damaged HSCs, in order to open up new therapeutic perspectives in the cure of blood malignancies.

## 5. Autophagy in Hematological Malignancies

Autophagy has different functional effects in hematological malignancies that seem to depend on cancer ontogenesis and type. Table 1 summarizes the current knowledge of autophagy in each specific hematological malignancy.

### 5.1. Autophagy and Multiple Myeloma

MM is the second most common hematologic neoplasm [76]. It is characterized by a malignant proliferation of monoclonal plasma cells (PCs) in the bone marrow that synthesize huge amounts of immunoglobulins [77,78,79]. This causes a large production of misfolded proteins in the ER, which are potentially toxic to the cell. To avoid cytotoxicity, mechanisms such as the proteasome system, and chaperones, the response system to misfolded proteins and autophagy, are activated [80,81,82,83]. In addition, the rapid proliferation of MM cells and the synthesis of immunoglobulins demand high energy requirements, which are in part supplied by autophagy, degrading excess proteins to obtain energy [81]. Therefore, the baseline levels of autophagy in MM are high compared to in other tumors, as it is an essential process for tumor survival [84]. Autophagy would fulfill two main functions: the elimination of ubiquitinated proteins in coordination with the UPS, and the containment of the secretory system, as occurs in normal PCs [85]. Thus, it constitutes an important therapeutic target that could either eliminate the protective effect on the tumor cell when inhibited, by increasing the number of misfolded proteins and limiting energy supply; alternatively, it could be increased in a way that promotes autophagy-mediated cell death.

There is evidence for the proapoptotic role of autophagic inhibition. Betulinic acid is a cyclic triterpene with antitumor properties that inhibits autophagic flux in MM cells and induces apoptosis, which was verified by the accumulation of LC3-II and p62 and high expression of caspase-3 [86]. Genetic inhibition of autophagy has similar effects. Performing knockdown of Linc00515 (a long noncoding RNA) and ATG14 decreases autophagy and chemoresistance to melphalan, an alkylating agent [87]. On the other hand, blocking the adapter protein SQSTM1/p62, which plays a key role in selective autophagy, causes failure in loading of the autophagosomal content and, therefore, inefficient autophagy, which leads to apoptosis [88]. However, an exacerbated activation of autophagy can also cause cell death, as seen with metformin treatment, which induces autophagy and cell-cycle arrest in the G0/G1 phase through the activation of AMPK and repression of mTOR [89]. Similarly, NVP-BEZ235, a new class I PI3k/mTOR inhibitor, efficiently inhibits cell proliferation in MM cells and induces apoptosis and autophagy [90]. Another example is asiatoside, a triterpenoid with antitumor effects in MM that inhibits the growth of cells resistant to BTZ, in part due to the induction of autophagy, defined as LC3-II overexpression [91].

Standard first-line treatment for the treatment of MM includes PIs, as well as glucocorticoids, immunomodulators, HDAC inhibitors, and monoclonal antibodies. MM survival rates have been increasing over the years, but many patients relapse due to drug resistance [92]. Autophagy is thought to be involved in the resistance to BTZ, which affects around 15–20% of MM patients [51]. For this reason, combined treatments focused on simultaneously blocking the proteasome and autophagy pathways seem to be an appropriate solution to overcome this resistance. Clinical trials in MM patients treated with BTZ and HCQ showed that the combination is well tolerated and, to some extent, improves the disease outcome [93]. Both drugs act by increasing the ER stress level due to increased accumulation of misfolded toxic immunoglobulins, which predispose the cell to undergo apoptosis [51]. A similar response occurs when combining carfilzomib (a second-generation PI) and CQ/HCQ, enhancing cellular apoptosis in MM cells [94], which is also achieved by inducing autophagy and inhibiting aggresome formation using HDAC6 inhibitors, such as ACY-121561 [95]. Lastly, recent evidence has also suggested that higher immunoreactivity for autophagy proteins such as Beclin-1 and L3C is correlated with better disease outcomes [96]. 

### 5.2. Autophagy and Leukemia

#### 5.2.1. The Role of Autophagy in Chronic Lymphocytic Leukemia 

Chronic lymphocytic leukemia (CLL) is the most common type of leukemia in developed countries, mainly affecting elderly people [97]. It is characterized by the proliferation and accumulation of morphologically mature but immunologically dysfunctional B cells in the blood, bone marrow, lymph nodes, and spleen, which is sometimes preceded by an oligoclonal expansion of B lymphocytes, called lymphocytosis [98]. Apoptosis and necroptosis are the two main processes of cell death, but numerous studies suggest that autophagy can mediate both death and cell survival processes in CLL, constituting a potential therapeutic target [14,32]. As main cell death pathways are linked, it is important to understand the molecular mechanisms interconnecting them to choose the appropriate drug combination for each case.

Some studies have suggested a role for the autophagy process in promoting cytotoxicity and apoptosis of tumor cells [99]. Recently, autophagic flux in peripheral blood mononuclear cells from CLL patients was measured by Western blot of p62 and LC3-II. Both lymphocytosis and a high percentage of tumor lymphocytes were associated with blocked autophagic flow, which could be one of the first alterations associated with CLL [100]. This is consistent with previous studies reporting that autophagy inhibition mediated by MGCD0103, a histone deacetylase inhibitor that acts through the PI3K/AKT/mTOR and CAPN1 pathways, promoted CLL cell death [101]. In addition, the Bcl-2 family of proteins decreases autophagy and, thus, reduces the therapeutic effects of the drugs used to treat CLL. In patients resistant to treatment with fludarabine, resistance generally occurs when the expression of members of the Bcl-2 family are enhanced [102]. Mcl-1 associated with Beclin-1 inhibits cell death in fludarabine-resistant cells. Furthermore, in sensitive cells, the inhibition of autophagy by siBeclin-1, shATG7, shLAMP2, and the 3-MA inhibitor significantly reduced the death of CLL cells [103]. However, applying obatoclax promoted the dissociation of Beclin-1 and Mcl-1, activating autophagy and producing high cytotoxicity, suggesting that Beclin-1-dependent autophagy is an effective mechanism to overcome resistance to fludarabine [103]. Other drugs, such as venetoclax or flavopiridol, have also been shown to favor cell death in CLL by activating autophagy [104]; however, tumor lysis syndrome was observed in 40% of patients [105]. In subsequent studies attempting to reduce toxicity, cyclophosphamide, flavopiridol, and rituximab were combined, obtaining favorable results [106]. Furthermore, it has been reported that the SLAMF1 protein activates autophagy by indirectly stabilizing the Beclin-1–VPS34 complex, and that SLAMF1-cells are less sensitive to autophagy-inducing therapies [107]. Kong et al. found higher levels of Beclin-1 and ATG5 mRNA in CLL patients compared to healthy individuals [108], whereas Arroyo et al. observed that higher concomitant expression of LC3B, CD38, and ZAP70 proteins was correlated with faster disease progression [109]. The above background suggests that the role of autophagy in CLL may differ due to the enormous disease heterogeneity; therefore, appropriate context must be given. 

On the other hand, numerous studies point to autophagy as a mechanism of cytoprotection and survival of the tumor cells, making them resistant to conventional drug treatments [110,111]. When treating CLL patients with dasatinib (a tyrosine kinase inhibitor), the drug triggers stress in the ER and reduces the expression of Mcl-1 and Bcl-xL, increasing cytoprotective autophagy and causing resistance [110]. This suggests that the combination of antitumor drugs with autophagy inhibitors could improve therapy outcomes in CLL patients [110,112]. Autophagy can also protect the tumor when activated, not only in tumor cells, but also in stromal cells from the niche surrounding the tumor. This mechanism has been described in resistance to vorinostat, an HDAC inhibitor. By using the autophagy inhibitors 3-MA and CQ, it is possible to recover cells’ sensitivity to vorinostat [113]. Other drugs, such as MGCD013, act by decreasing autophagic flux directly in CLL cells, thereby enhancing cell death induced by inhibitors of HDACs and cyclin-dependent kinases (CDKs) [101].

#### 5.2.2. Role of Autophagy in Acute Lymphocytic Leukemia

Acute lymphocytic leukemia (ALL) is divided into two classes: B-cell acute lymphocytic leukemia (B-ALL) and T-cell acute lymphocytic leukemia (T-ALL). They constitute 20% of acute leukemia in adults and are the most common type of hematologic malignancy in children [78]. B-ALL constitutes approximately 80% of cases and T-ALL the remaining 20% [79]. ALL is a highly heterogeneous disease, displaying a variety of genomic and chromosomic alterations including aneuploidies, translocations, alterations in the number of copies of DNA fragments, and mutations. T-ALL is an aggressive variant characterized by the uncontrolled proliferation of T-type lymphoblasts that arise in the thymus and express markers for immature T-lymphocytes, which accumulate in the blood [114]. B-ALL is characterized by the production of B-lymphocyte precursor cells that are produced in the marrow and accumulate in the blood. An important characteristic of B-ALL is the constitutive activation of the PI3K/Akt/mTORC1 pathway, which directly affects autophagy, although its specific role is not entirely well defined [115,116].

Treatment with glucocorticoids (GCs) is one of the most used in B-ALL; however, in many cases resistance occurs, causing poor prognosis [117]. Some authors have investigated the role of autophagy-related genes in drug resistance to GCs, showing that 36 genes were significantly expressed, 10 of them upregulated and 26 downregulated [118]. These results suggest that the resistance to GC may be modulated by the repression of autophagy key inducers and induction of key inhibitors [118]. In line with this hypothesis, Bonapace et al. found that induction of autophagy activated necroptosis, required by ALL-B cells to overcome resistance to GCs [119]. In addition, the processing of LC3 has been observed in cells treated with dexamethasone and a MEK inhibitor, showing that only the combination of drugs stimulates autophagy, increasing the toxicity of dexamethasone in resistant B-ALL cells and causing apoptosis [120]. Another solution to overcome resistance to GCs is the use of BTZ in childhood B-ALL cells [121]. Paradoxically, in cases where resistance to BTZ is developed, inhibition of autophagy helps to improve the sensitivity of the cells and enhances the effect of the drug [122]. Similarly, other studies have demonstrated the cytoprotective effect of autophagy, favoring resistance to multiple drugs. This is the case with resistance to L-asparaginase, used in the treatment of ALL for decades [123], but also with drugs such as 20(S)-ginsenoside Rh2, where, by inhibiting autophagy both pharmacologically and genetically, cellular viability is reduced through promoting drug toxicity [124].

In the case of T-ALL, hardly any studies have suggested that autophagy is an antitumoral mechanism. Thymosaponin A-III, an antitumor agent used in ALL, induces autophagy by inhibiting the PI3K/Akt/mTOR pathway and inhibits cell viability in a dose-dependent manner, inducing the expression of Beclin-1 and LC3-II, causing cell death via both autophagy and apoptosis [125]. A similar example is NVP-BKM120, a PI3K inhibitor that exhibits antitumor properties through the activation of apoptosis, as well as autophagy-mediated cell death [126].

Nevertheless, there is much more evidence that autophagy exhibits protective properties, aiding cell viability and chemoresistance. Using BTZ and obatoclax together in T-ALL, the blocking of protein degradation pathways is achieved, both UPS and autophagic, causing an accumulation of polyubiquitinated proteins and, therefore, increasing cytotoxicity [127]. The effect of autophagy on vorinostat resistance has also been demonstrated, as it occurs in CLL. Concomitant treatment with quinacrine (Qn), an antimalarial drug that inhibits autophagy, synergistically promoted cell death by vorinostat at a nontoxic concentration [128]. The inhibition of autophagy could thus constitute an antitumor mechanism against T-ALL, although it largely depends on the drug used. An example of this is the specific case of dihydroceramides C22:0 and C24:0, in which, although autophagy occurred when treating T-ALL cell lines, autophagic inhibition with 3-MA had no effect on cytotoxicity [129].

#### 5.2.3. Role of Autophagy in Chronic Myeloid Leukemia

Chronic myeloid leukemia (CML) is characterized by clonal myeloproliferation arising from transformed HSCs presenting the chimeric oncogene BCR-ABL [79]. The expression of BCR-ABL results from the translocation between chromosome 9 and 22, called the Philadelphia chromosome. This fusion protein is a constitutively active tyrosine kinase that initiates a series of metabolic pathways that lead to leukemogenesis. For this reason, the star treatment of CML is through tyrosine kinase inhibitors (TKIs) [130]. The specific process of autophagic regulation by BCR-ABL has not yet been established. Although it has been suggested that BCR-ABL suppresses autophagy by regulating the transcription of mTOR [131], there is also evidence that BCR-ABL promotes autophagic machinery. One of the ways in which BCR-ABL regulates autophagy is through the MAPK15 kinase, interacting with proteins of the LC3 family [132]. In fact, BCR-ABL^+^ cells appear to be highly dependent on autophagy, since apoptosis occurs rapidly when it is blocked by deleting ATG3 or inhibited with 3-MA and CQ [133].

What is clear is the key role of autophagy in resistance to TKIs that act against BCR-ABL [130]. CML is resistant to TKIs in approximately 13% of patients [134]. Imatinib and ponatinib, first- and third-generation TKIs, respectively, induce cytotoxicity and apoptosis, as well as trigger autophagy in resistant cells, which is associated with a cytoprotective effect and decreased cell death [135]. Second-generation TKIs, such as nilotinib and dasatinib, show similar behavior [130]. Numerous studies have attempted to restore sensitivity to tumor cells by inhibiting autophagy and increasing the effect of TKIs, with noticeable results. For instance, ponatinib efficacy increases when it is used in combination with HCQ or upon ATG7 knockdown, which inhibits both mTOR and the autophagic machinery [136]. HCQ is the only autophagy inhibitor currently approved by the FDA (Food and Drug Administration), but it does not consistently inhibit autophagy; therefore, two second-generation inhibitors have recently been tested: Lys05 and PIK-III, a lysosomotropic agent and a selective VPS34 inhibitor, respectively. The number of primary CML cells was reduced, blocking autophagy and promoting cell death [137]. Autophagy is also involved in resistance to other antitumor agents, such as diosgenin, which promotes drug-mediated cell death and autophagy in CML cells [138], or 20(S)-ginsenoside Rh2, whose induced autophagy may protect K562 and U937 cells from apoptosis [139].

#### 5.2.4. Role of Autophagy in Acute Myeloid Leukemia

Acute myeloid leukemia (AML) is characterized by dysregulation of cell proliferation and accumulation of immature myeloid progenitors in the bone marrow and blood. Although life expectancy has improved in young patients, in most adults, AML is associated with a poor prognosis, and many patients show drug resistance and subsequent relapse [116,140].

The most common genetic abnormality, present in around 30% of cases, is the alteration of the FMS-type tyrosine kinase 3 gene (FLT3), either in the form of point mutations or internal tandem duplication (ITD), which have been associated with a poor prognosis [141]. Therefore, treatments with TKIs and strategies modulating the expression of FLT3-ITD constitute promising approaches [142,143]. Furthermore, AML samples with this type of mutation have been shown to be more sensitive to proteasome inhibitors (PIs) [144]. PIs have been shown to be capable of overcoming resistance to quizartinib (a second-generation ITQ), induced by mutations in the kinase domain of FLT3, suggesting that the combined application of both drugs may produce favorable results. PIs act by activating cytotoxic autophagy, which degrades FLT3-ITD in autophagosomes, as was verified with treatment with BTZ, such that autophagy would play an antitumor role [144]. Another study affirmed that the expression of FLT3-ITD increases basal autophagy in AML cells through a mechanism involving the ATF4 transcription factor [145]. In addition, via the use of shRNAs that reduce the expression of key autophagy proteins, it was shown that autophagy is required for the proliferation of AML cells in a murine model, giving autophagy a protumorigenic role and showing that its inhibition exceeds the resistance to the FLT3 inhibitor quizartinib [145].

In the de novo AML, the baseline autophagy level seems to be low. The expression of the ATG7 and LC3 genes has been studied, and both showed a decrease in AML patients compared to healthy controls, showing a positive correlation between ATG7 and LC3, whereby a reduction in autophagy genes could lead to the initiation of leukemogenesis [146]. These findings are in line with a previous study by Jin et al. that demonstrated low expression of the ULK1, ATG3, ATG4D, and/or ATG5 genes in AML patients, partially due to inhibition of its positive regulator, PU.1 [147]. In addition, this study reported that their expression could be restored by all-trans retinoic acid (ATRA) therapy in AML cell lines that initiate autophagy via the Beclin-1 pathway [147]. Likewise, the antitumor alkaloid matrine inhibits colony formation by inducing apoptosis and autophagy in AML primary cell lines. It is known that matrine blocks the phosphorylation of Akt, mTOR, and their substrates, leading to autophagy promotion [148]. In addition, the level of LC3-II is positively regulated and p62 expression is reduced, indicating that autophagy is involved in the anti-AML effect [148]. Thus, bafilomycin-A, an autophagy inhibitor, reduces cytotoxicity, while rapamycin activation has the opposite effect [149]. Another autophagy inducer is typhaneoside (TYP), a flavonoid drug that reduces cell viability in AML cells through cell cycle arrest and apoptosis. This drug clearly induces ferroptosis, associated with autophagy, by promoting AMPK activation, which contributes to ferritin degradation, ROS accumulation, and ultimately ferroptotic cell death [150]. However, other studies associate the induction of autophagy with a poor prognosis of AML due to its cytoprotective effect [151,152]. It has been reported that the expression of the surface marker CXCR4 in AML is associated with increased autophagic flux and decreased apoptosis, mediated by the antitumor drug cytarabine [153]. Therefore, the combined use of autophagy inhibitors might significantly increase cellular sensitivity to the drug. A similar example is JL1037, a specific lysine demethylase 1 (LSD1) inhibitor that induces apoptosis and autophagy in AML. Furthermore, cotreatment with CQ enhanced the effect of JL1037 by promoting both processes [154].

### 5.3. Autophagy and Lymphomas

Lymphomas, hematological malignancies that develop in the lymphoid system, are one of the most common forms of cancer overall, with growing incidence in the last few decades [155]. There are two main types of lymphoma, with different therapeutic management indicated, Hodgkin’s lymphoma and non-Hodgkin’s lymphoma, encompassing around 40 subtypes [156]. Treatment options include chemotherapy, radiation therapy, and monoclonal antibodies, but many of them have pronounced side-effects, such as infertility, cardiac diseases, or secondary cancers, while chemotherapy resistance and disease relapse remain issues. In the search for novel therapeutic targets to tackle these challenges and improve prognosis, new signaling pathways and specific inhibition approaches are being investigated. In solid tumors, autophagy protects cells from hypoxia and nutrient deprivation and prevents apoptosis; therefore, autophagy-based strategies represent a new approach to clinical treatment and an opportunity to improve the current results.

### 5.4. Autophagy and Hodgkin’s Lymphoma

Hodgkin’s lymphoma (HL) accounts for approximately 10% of all lymphomas [157]. The hallmark cells of HL are the Hodgkin and Reed–Sternberg (H-RS) cells, whose origin remains unknown, although Epstein–Barr virus is thought to be partially responsible. The WHO has established two clinical and biologically different types of HL: classic HL and nodular lymphocyte-predominant HL [79]. This is characterized by the low presence of malignant cells in the tumor and the abundance of accompanying nonmalignant reactive cells, such as lymphocytes, eosinophils, neutrophils, histiocytes, and plasma cells [158]. 

Most of the information on autophagy in HL comes from gene expression profiling. Birkenmeier et al. studied autophagy basal levels in classic HL (cHL), following the premise that autophagy is involved in both tumor suppression and progression. They found constitutive autophagy activation in cHL cells and primary tissue from patients [159]. The expression of key proteins such as Beclin-1, ULK1, and LC3 was increased in cHL. In addition, cells showed an elevated number of autophagic vacuoles and an intact autophagic flux. Autophagy inhibition by CQ or ATG5 inactivation induces apoptosis and impairs cHL cell proliferation. In mice, CQ inhibition of basal autophagy also significantly curbs HL growth [159]. This was in line with other previous studies, in which autophagy activation was triggered in response to apoptosis activators, and its inhibition with either CQ or ATG5 short hairpin RNA (shRNA) induced tumor cell death in a Myc-induced model of lymphoma [160]. These promising results may suggest autophagy inhibition strategies as an appropriate therapeutic option; however, there is also evidence showing autophagy activation leading to tumor cell death. Estrogens, particularly 17β-estradiol (E2), may play a key role in HL through estrogen receptors (ER) β activation. DPN (2,3-bis(4-hydroxyphenyl)-propionitrile) is a selective agonist that binds ERβ, activating it and leading to a reduction in in vitro cell proliferation and cell-cycle progression by inducing autophagy. An overexpression of the damage-regulated autophagy modulator 2 (DRAM2) molecule was found, and, after ERβ activation, both DRAM2 and LC3 interacted with each other and were located at a mitochondrial level. These results suggest that autophagy may play an important role in tumor growth suppression [161]. In fact, the histone deacetylase (HDAC) inhibitor LBH589 (panobinostat) has shown clinical efficacy in H-RS cells in HL through its impact on the microenvironment, allowing lymphocyte activity. In HL cell lines, it was observed that LBH589 caused cell death and autophagy, and it may further promote an increased susceptibility against effector cell killing [162]. Recently, melatonin has been found to exert antitumor activities in HL, reducing cell proliferation and promoting cell apoptosis. HL cells treated with melatonin showed increased expression of LC3-II and decreased p62 proteins, with an enhanced production of autolysosome and activation of autophagy [163]. The expression of other proteins such as G protein-coupled receptors MT2 and retinoic acid-related orphan receptors (RORs) was also increased. RORC, a type of ROR, is thought to induce autophagy activation when overexpressed. Therefore, melatonin exhibits tumor-suppressive effects due to an increased level of RORC-induced autophagy [163].

### 5.5. Autophagy and Non-Hodgkin’s Lymphoma

Non-Hodgkin’s lymphoma (NHL) is a group of blood cancers that includes all types of lymphomas except HL. The classification of NHL by the WHO depends on the type of lymphocytes that turn malignant, as well as their morphology, immunohistochemistry, chromosome features, and surface markers [164]. Generally, NHLs are divided into B-cell lymphomas and T-cell lymphomas, each with many different subtypes [165]. 

#### 5.5.1. Autophagy and B-Cell Lymphoma

Upregulated-autophagy is associated with a favorable clinical outcome in NHL, as expression of Beclin-1, an oncosuppressor, is upregulated in B and T lymphocytes. NHLs with >20% of tumor cells expressing a high level of Beclin-1 aggregates have been associated with complete or partial remission and increased overall survival [166]. This suggests that NHLs with upregulated autophagy are more sensitive to chemotherapy. Furthermore, it was reported that Beclin-1-deficient mice had a high risk of developing spontaneous precursor B-cell lymphoma, evidencing a key role of Beclin-1 and autophagy in normal cell maintenance [167]. However, autophagy can act as a double-edged sword in both leukemia and lymphoma, and its specific role may depend on the disease subtype.

#### 5.5.2. Burkitt’s Lymphoma

Burkitt’s lymphoma is a highly aggressive type of NHL that originates from the follicle germinal center and sometimes is manifested as acute leukemia. Over time, great efforts have been made to understand the pathology of Burkitt’s lymphoma and develop new and effective treatments, as well as elucidate the mechanisms of chemotherapy-induced cell death including autophagy [168]. There are several studies defending the antitumor role of autophagy in Burkitt’s lymphoma. It has been reported that BAFF, a B-cell-activating factor from the TNF family, controls and inhibits autophagy, thereby contributing to cell proliferation and survival by activating the Akt/mTOR signaling pathway in Raji cells [169]. This suggests that promoting autophagy may be a strategy for the prevention of excessive BAFF-induced aggressive B lymphocyte disorders. Moreover, the induction of autophagy is related to antitumor outcomes. Drugs that had cytotoxic effects in other tumors, such as artesunate [168] or ouabain [170], induced not only apoptosis but also autophagy in Raji cells. Both treatments decrease cell proliferation and promote cell death by inducing both apoptosis and autophagy. Furthermore, induction of autophagy can improve the cytotoxic effects of some drugs. chLym-1 is a chimeric antihuman HLA-DR monoclonal antibody used to treat cancer, which also induces autophagy in Raji NHL cells [171]. Inhibition of autophagy by 3-MA or NH4Cl or genetic approaches suppresses chLym-1-induced growth inhibition, apoptosis, and cytotoxicity, while induction of autophagy accelerates their antitumor effects, suggesting that autophagy plays an important role in the tumor-suppressing effects of the chLym-1 antibody [171]. Similar results were found with the antibody drug conjugate (ADC) rituximab–monomethyl auristatin E (MMAE), which triggered apoptosis and autophagy [172]. Autophagy inhibition by CQ suppressed rituximab–MMAE-induced apoptosis, whereas activating autophagy enhanced the anticancer effect of the drug both in vitro and in vivo [172]. Moreover, other treatments such as phototherapy with a blue light-emitting diode also induce apoptosis and autophagy, which in fact contribute to cell apoptosis. The formation of autophagosomes and level of LC3-II were increased in Burkitt’s lymphoma A20 and RAMOS cells exposed to a blue LED, and inhibition of autophagy by 3-MA reduced the apoptosis activation, evidencing a key role of autophagy in the antitumor effect of this treatment [173]. Another antitumor mechanism in which autophagy is involved is autophagic cell death. The HDAC inhibitor valproic acid (VPA), combined with the mTOR inhibitor temsirolimus, inhibited cell growth and triggered autophagic cell death in Burkitt’s lymphoma cell lines, primary tumor cells, and a murine xenograft model, reducing tumor growth [174].

Autophagy is also associated with a cytoprotective, protumor role, favoring drug resistance. Inhibiting autophagy by 3-MA helped overcome Raji cell resistance to rapamycin, which was tested as a potential GC sensitizer in Burkitt’s lymphoma cells [175]. Similar results were found for vismodegib, an inhibitor of the Hedgehog signaling pathway that induced cytotoxicity and apoptosis in Raji cells, as well as activated autophagy [176]. On the other hand, autophagy inhibition potentiated cytotoxicity and apoptosis induced by vismodegib and accelerated the formation of ROS, being able to potently kill Raji cells and overcome vismodegib resistance [176]. Inhibition of autophagy using pharmacological (3-MA) or genetic approaches (siRNA of ATG5 and Beclin-1) has also been used to improve the antitumor effects of recombinant human arginase (rhArg), which induced growth inhibition, cell-cycle arrest, and caspase-dependent apoptosis in Raji and Daudi cells through arginine deprivation [177].

#### 5.5.3. Mantle Cell Lymphoma

Mantle cell lymphoma (MCL) is an uncommon and invasive subtype of non-Hodgkin’s lymphoma, with a poor prognosis, derived from mature B cells that often do not enter the follicular germinal center [178] and that often present aberrantly high cyclin D1-driven CDK4 activity. The high proportion of chemoresistance in MCL patients makes it important to elucidate the mechanisms involved, such as the autophagy pathway, which can act as a cytoprotective mechanism in MCL. In fact, transglutaminase TG2 has been linked to constitutive activation of NF-κB and chemotherapy resistance in MCL cells [179]. TG2 binds NF-κB components to form complexes, as well as plays an unexpected role, triggering autophagy in drug-resistant MCL cells through the induction of IL6 [179]. Under stress conditions, TG2 and IL6 mediate enhanced autophagy formation to promote cell survival. Interestingly, ATG5 positively regulates TG2/NF-κB/IL6 signaling, suggesting a positive feedback loop, whose disruption may constitute a good therapeutic strategy to overcome drug resistance in MCL [179]. Moreover, the blockage of autophagy enhances the anticancer efficacy of several drugs. An example of this is everolimus, an inhibitor of rapamycin (mTOR) kinase that has shown activity in preclinical and clinical models of MCL, but an accumulation of autophagic vacuoles is correlated with a lack of efficacy of the treatment. The complete therapeutic potential could be restored by ATG gene selective knockdown or pharmacological inhibition by HCQ, overcoming the resistance [180]. Another example is FTY720, a synthetic sphingosine analog with autophagy-blocking and antineoplastic effects. In MCL, FTY720 enhanced the pro-death activity of milatuzumab, a humanized monoclonal antibody, by inhibiting the autophagy–lysosome-dependent degradation of its therapeutic target, CD74, suggesting a potential therapeutic use in combined therapy [181]. Inhibitors of the UPS have shown promising results, but stabilization of the short-lived proapoptotic NOXA is a critical determinant of the sensitivity to these inhibitors. Combined treatment with BTZ and autophagy inhibitors enhanced NOXA stability, leading to superinduction of the NOXA protein. This combination induced apoptosis in both MCL cell lines and patient samples, causing cell death and suggesting that inhibitors of autophagy are promising candidates to increase the activity of proteasome inhibitors in MCL [182]. 

However, there is evidence that autophagy can also play an antitumor role. Temsirolimus is an inhibitor of the mTOR pathway. MCL cells treated with temsirolimus showed mTOR inhibition and cell-cycle arrest, together with an increased number of acidic vesicular organelles and LC3-I levels, indicating activation of autophagy. When MCL cells were treated with vorinostat, it activated caspase 3 and induced apoptosis, suggesting that a combination of the drugs could have synergistic antiproliferative effects, targeting both apoptosis and autophagy [183]. These results are evidence of the need to study whether autophagy causes cell death or improves survival in MCL.

#### 5.5.4. Primary Effusion Lymphoma

Primary effusion lymphoma (PEL) is a subtype of NHL, located in the body cavities, characterized by pleural, peritoneal, and pericardial fluid lymphomatous effusions without detectable tumor masses. It accounts for approximately 4% of HIV-associated NHL. PEL is always associated with human herpes virus-8 and shows an extremely aggressive clinical course, with a poor prognosis of patients even under conventional chemotherapy, hence the need for a novel therapeutic strategy against PEL [184,185]. In this hematological malignancy, autophagy predominantly serves as a cell survival mechanism; therefore, its inhibition helps with overcoming drug resistance and provokes tumor cell death. Treatment of PEL cells with CQ inhibited autophagy and induced caspase-dependent apoptosis in vitro, suppressing cell growth. CQ activated ER stress signal pathways; thus, the inhibition of autophagy induces ER stress-mediated apoptosis in PEL cells, constituting a novel strategy for cancer chemotherapy [186]. Moreover, inhibition of autophagy also enhances the chemosensitivity of drugs. PEL cells constitutively activate STAT3 to survive; thus, tyrosine kinase inhibitors, such as AG490, are used to cause apoptotic cell death. AG490 was found to induce autophagy, whose inhibition by bafilomycin A or by Beclin-1 silencing increased its cytotoxic effect against PEL cells, evidencing the cytoprotective role of autophagy in PEL cells [187]. Another drug used to treat PEL cells that induced autophagy activation was quercetin, a bioflavonoid with anti-inflammatory and anticancer properties. Quercetin inhibited the PI3K/Akt/mTOR and STAT3 pathways in PEL cells, reducing the survival of the tumor cells and leading to cell death. Inhibition through Beclin-1, silencing the prosurvival autophagy in these cells, increased the cytotoxic effect of quercetin, suggesting that, in combination with autophagy inhibitors, quercetin may represent a feasible candidate for the treatment of PEL [188]. BTZ, used to treat MM, also induces apoptosis in PEL cell lines, promoting ER stress and activating the unfolded protein response, and triggering apoptosis and autophagy, which has a prosurvival role. The combination of BTZ with autophagy inhibitors such as ATG5 knockdown or pharmacological autophagy inhibitors could improve the outcome of the therapy for this aggressive B cell lymphoma [189].

#### 5.5.5. Diffuse Large B-Cell Lymphoma 

Diffuse large B-cell lymphoma (DLBCL) is an aggressive, clinically heterogeneous tumor with an incidence rate of 7–8 cases per 100,000 people per year, being the most common subtype of NHL [79]. It accounts for about 30–40% of all human lymphoid malignancy and occurs most frequently in elderly patients [190]. Although aggressive chemotherapy is the main choice for the treatment of this disease, the success of the treatment is often hampered by drug resistance [191]. There is evidence that autophagy can act as an antitumor mechanism in DLBCL. This disease is dependent on mitochondrial lysine deacetylase SIRT3 for proliferation and survival. In fact, SIRT3 knockout attenuates B-cell lymphomagenesis, impairs glutamine flux to the TCA cycle, and induces cell death and autophagy [192]. Interestingly, it was observed that SIRT3 inhibited autophagy in DLBCL cells, acting as a tumor-suppressive mechanism. In addition, ATG5 knockdown impaired autophagy in DLBCL and caused resistance to SIRT3 knockdown. Hence, the proliferation arrest induced by SIRT3 depletion in DLBCL is, at least in part, caused by autophagy, which is triggered by impairment of the TCA cycle [192]. Another example of autophagy improving prognosis is the inhibition of lncRNA MALAT-1. Metastasis-associated lung adenocarcinoma transcript-1 (MALT-1) is a highly conserved lncRNA that constitutes a hotspot in human cancer, whose role regulating autophagy signaling pathway in DLBCL is still poorly understood. In mice, when MALAT-1 was silenced by siRNA-MALAT-1, the tumor significantly reduced its volume and weight, and the formation of autophagosomes was observed. LC3-II/LC3-I and ATG5 expression was increased, and p62 expression was decreased when compared to the blank group. This suggests that inhibiting lncRNA MALAT-1 can improve the sensitivity to chemotherapy of DLBCL by enhancing autophagy-related proteins [193]. Furthermore, it is known that, in DLBCL, oncogenic MYC could induce choline metabolism by transcriptionally activating the key enzyme PCYT1A, resulting in impeded mitophagy-dependent necroptosis of the lymphoma cells [194]. This mechanism can be restored by BBR, a lipid-lowering alkaloid berberine, which showed anti-lymphoma activity, inducing mitophagy-dependent necroptosis. These results demonstrate a key antitumor role of mitophagy against DLBCL [194]. 

However, there is also evidence that autophagy may act as a cell survival mechanism. It was found that CUL4B, whose aberrant expression is related to various types of cancers, was highly expressed in both DLBCL tissues and cells lines, as well as related to a poor prognosis of the disease [195]. In addition, CUL4B acted as a potent inductor of autophagy through JNK phosphorylation. Its silencing caused inhibition of proliferation activity, which may be attributed to the blocking of the prosurvival ability, mediated by autophagy [195]. 

#### 5.5.6. Follicular Lymphoma

Follicular lymphoma (FL) is the second most common lymphoma diagnosed in the USA and Western Europe, accounting for approximately 20% of NHLs, with considerable variation by race/ethnicity and geography [196]. Tumorigenesis starts in precursors of specific types of B-cells such as centrocytes and centroblasts of the germinal center in the follicle. The gold standard treatment for FL includes R-CHOP (rituximab, cyclophosphamide, doxorubicin, vincristine, and prednisone) and R-bendamustine protocols. These drugs are administered either alone or in combination with chemotherapy, showing largely favorable outcomes. However, some patients can relapse, which makes FL an indolent, but still incurable malignancy that frequently evolves into an aggressive subtype resembling germinal center-derived DLBCL, with poor clinical outcomes [197]. FL and some cases of DLBCL overexpress the antiapoptotic protein BLC-2. A great number of autophagy-related genes were found to be upregulated in FL and DLBCL cell lines and in primary cells from malignant human lymph node biopsies, showing seven autophagy genes upregulated in FL B-cells and only two genes in DLBCL B-cells, and demonstrating that FL had increased basal autophagy activity regardless of overexpression of BCL-2 [198]. Moreover, LC3A expression was increased in FLs when compared to DLBCL and was induced by hypoxia (related to HIFα) [199]. Together, these results point to autophagy as a potential therapeutic target. In fact, mutations in the human vacuolar ATPase subunit ATP6V1B2 activate autophagic flux and maintain mTOR in an activate state. Primary FLB cells carrying mutated ATP6V1B2 showed a remarkable ability to survive in low leucine concentrations. Furthermore, treatment with inhibitors of autophagy such as bafilomycin A was successful, with the majority of primary FL B cells carrying ATP6V1B2 mutations being preferentially sensitive to the treatment, unlike WT ATP6V1B2 B cells [200].

#### 5.5.7. Autophagy and T-Cell Lymphoma

Given that T-cell lymphoma is a relatively rare NHL [201,202], it has been challenging to identify subtype-specific risk factors. There are two main subtypes of T-cell NHL: peripheral T-cell lymphoma (PTCL) and cutaneous T-cell lymphoma (CTCL) [202], which has much lower prevalence. The therapeutic management includes R-CHOP14, DA-EPOCH, and GemDox protocols, as well as HDAC inhibitors, immunoconjugates, CD52 monoclonal antibody, and folic acid antagonists [203]. Despite all the options available, the therapeutic outcome of T-cell lymphoma patients is still poor; hence, there is an unmet need for novel treatment strategies. To date, it has been complicated to study the link between autophagy and T-NHL.

##### Cutaneous Cell Lymphoma

As explained before, several studies have demonstrated the important role of autophagy in the malignancy of lymphoma. In cutaneous T-cell lymphoma, Yan et al. described how hypoxia-induced autophagy lessens the sensitivity of Hut78 cells to doxorubicin, thereby inducing chemoresistance. Moreover, when autophagy was inhibited by 3-MA, the apoptosis rate increased and the cell viability was attenuated [204]. These results point to autophagy as a potential target against drug resistance in T-NHL. In the same vein, Zhang et al. studied the role of miR-449 in the regulation of tumorigenesis and autophagy, discovering that it enhanced apoptosis of T-cell lymphoma by decreasing the degree of autophagy [203]. Another example of drug resistance being mediated by autophagy is sinensetin, a methoxyflavonoid with anticancer effects that induced cell death, apoptosis, and autophagy in Jurkat cells. Inhibition of autophagy by 3-MA significantly enhanced the apoptosis rate and improved the sensitivity of the Jurkat cells to sinensetin, suggesting that cotreatment with an autophagy inhibitor and sinensetin could be a potential candidate strategy to treat T-cell lymphoma [205].

##### Peripheral T-Cell Lymphoma

In peripheral T-cell lymphoma, there is also evidence that autophagy is associated with the malignancy of lymphoma. For instance, Wang et al. demonstrated that BCYRN1, a c-MYC-activated long noncoding RNA, could induce resistance to asparaginase by inducing autophagy in extranodal NK/T-cell lymphoma (ENKTCL) [206]. Thus, this finding suggested that pharmacological autophagy inhibition could be beneficial to restore the asparagine sensitivity of ENKTCL cells. Something similar happens with anaplastic large-cell lymphoma. Anaplastic large-cell lymphoma (ALCL) is a rare type of T-NHL that comprises about <5% of all NHLs [207]. It is an aggressive form of malignant lymphoma that affects mostly children and young adults [114]. There are two systemic forms of ALCL, depending on the presence of absence of aberrant anaplastic lymphoma kinase (ALK) expression, carrying translocations at the 2p23 region. In the case of ALK^+^ ALCL, although not often, patients can be treated with small-molecule inhibitors such as crizotinib. Recently, it has been suggested that autophagy can be involved in drug-resistant mechanisms for crizotinib. In fact, cells treated with crizotinib or ALK-targeting siRNA activated cytoprotective autophagy, suggesting that cotreatment with crizotinib and CQ could be beneficial for ALK^+^ ALCL patients [208]. Nevertheless, a recent study from the same group showed that crizotinib-mediated inactivation of ALK caused an increase in BCL2 levels that impeded the cytotoxic effects of the drug. BCL2 downregulation, together with crizotinib treatment, potentiated cell viability loss through an increase in autophagic flux and cell death. Furthermore, blockade of autophagic flux reversed impaired cell viability. This may suggest that exacerbated autophagy is associated with cell death, representing a promising therapeutic alternative to current ALK^+^ ALCL treatments [208]. As a matter of fact, activation of autophagy by AMPK activator metformin inhibited the growth of both B and T lymphoma cells, showing a strong antitumor effect. Lymphoma cells responded significantly better to doxorubicin and mTOR inhibitor temsirolimus when cotreated with metformin [209]. In fact, cotreatment of oral metformin with doxorubicin or temsirolimus triggered lymphoma cell autophagy and inhibited cell growth more efficiently than either agent alone, which seems to be contradictory to the abovementioned study by Yan et al., where hypoxia-induced autophagy lessened the sensitivity of lymphoma cells to doxorubicin and caused chemoresistance [204]. However, there is evidence that other drugs, such as fenugreek extract, can induce cellular death by activating autophagy-associated death in T lymphoma Jurkat cells [210]. 

Given all the above, it is important to highlight once again the dual role of autophagy in hematological malignancies and the importance of carefully investigating the specific molecular mechanisms that regulate autophagy in each specific malignancy.

**Table 1 cancers-14-05072-t001:** Description of autophagy findings in different hematological malignancies.

Disease	Therapy	Effect on Autophagy	Organism/Cell Types	Reference
MM	Betulinic acid	Inhibition	MM cell lines	[88]
	Melphalan	Inhibition	MM cell lines and MM cells resistant to melphalan	[89]
	Metformin	Activation	MM cell lines	[91]
	NVP-BEZ235	Activation	MM cell lines	[92]
	Asiatoside	Activation	MM cells resistant to BTZ	[93]
	BTZ and HCQ	Activation	MM cell lines	[95]
	Carfilzomib + CQ/HCQ	Activation	MM cell lines	[96]
	ACY-121561	Inhibition	MM cell lines and ANBL-6 BTZ sensitive and resistant cells	[97]
CLL	MGCD0103	Inhibition	CLL cell lines	[103]
	Obotoclax	Activation	Human pre-B acute lymphocytic leukemic cell lines and fludarabine-resistant cells	[105]
	Venetoclax	Activation	CLL cell lines	[106]
	Flavopiridol	Activation	CLL cell lines	[106]
	Cyclophosphamide + flavopiridol + rituximab	Activation	CLL cell lines	[108]
	Dasatinib	Activation	CLL cell lines	[112]
	3-MA and CQ (Vorinostat)	Inhibition	Primary CLL cells	[115]
ALL	Resistance to Glucocorticoids	Inhibition	B-ALL cell lines	[120]
	Dexamethasone and MEK inhibitor	Activation	B-ALL cell lines	[122]
	BTZ (Resistance)	Inhibition	B-ALL cell lines	[124]
	L-Asparaginase	Inhibition	B-ALL cell lines	[125]
	20(S)-Ginsenoside Rh2	Inhibition	B-ALL cell lines	[126]
	Thymosaponin A-III	Activation	T-ALL cell lines	[127]
	NVP-BKM120	Activation	T-ALL cell lines	[128]
	BTZ + Obotoclax	Inhibition	T-ALL cell lines	[129]
	Quinacrine	Inhibition	T-ALL cell lines	[130]
	Dihydroceramides C22:0 and C24:0	Inhibition	T-ALL cell lines	[131]
	3-MA	Inhibition	T-ALL cell lines	[131]
CML	3-MA and CQ	Inhibition	BCR-ABL^+^ cell lines	[135]
	Imatinib	Activation	BCR-ABL^+^ cell lines	[137]
	Ponatinib	Activation	BCR-ABL^+^ cell lines	[137]
	Nilotinib	Activation	BCR-ABL^+^ cell lines	[132]
	Dasatinib	Activation	BCR-ABL^+^ cell lines	[132]
	Ponatinib + HCQ	Inhibition	Primary CML cells	[138]
	Lys05	Inhibition	Primary CML cells	[139]
	PIK-III	Inhibition	Primary CML cells	[140]
	Diosgenin	Activation	CML cell lines	[140]
	20(S)-Ginsenoside rh2	Activation	K562 and U937 cells	[141]
AML	BTZ	Activation	FLT3-ITD	[146]
	shRNAs	Inhibition	AML cells (Murine model)	[147]
	Quizartinib	Inhibition	AML cells	[147]
	Trans-retinoic acid (ATRA) therapy	Activation	AML cell lines	[149]
	Alkaloid matrine	Activation	AML cell lines	[150]
	Bafilomycin-A	Inhibition	AML cells	[151]
	Rapamycin	Activation	AML cells	[151]
	Typhaneoside (TYP)	Activation	AML cells	[152]
	Cytarabine	Activation	AML cells	[155]
	JL1037	Activation	AML cells	[156]
	JL1037 + CQ	Activation	AML cells	[156]
HL	CQ	Inhibition	Murine model/Myc-induced model of lymphoma	[161]
	ATG5 short hairpin RNA (shRNA)	Inhibition	Myc-induced model of lymphoma	[162]
	DPN	Activation	HL cells (in vitro)	[163]
	LBH589 (panobinostat)	Activation	HL cell lines	[164]
	Melatonin	Activation	HL cell lines	[165]
NHL, Burkitt’s lymphoma	Artesunate	Activation	Raji cells	[170]
	Ouabain	Activation	Raji cells	[172]
	chLym-1	Activation	Raji cells	[173]
	Rituximab–monomethyl auristatin E	Activation	NHL cells	[174]
	CQ	Inhibition	HL cell lines	[174]
	Phototherapy	Inhibition	Raji Cells	[175]
	3-MA	Inhibition	Raji Cells	[175]
	Valproic acid + temsirolimus	Activation	Murine xenograft model	[176]
	Vismodegib	Activation/Inhibition	Raji cells	[178]
NHL, mantle cell lymphoma	Everolimus	Inhibition	Clinical models of MCL	[182]
	FTY720 + milatuzumab	Inhibition	MCL cell lines	[183]
	BTZ	Inhibition	MCL cell lines and patient samples	[184]
	Temsirolimus	Activation	MCL cell lines	[185]
	Vorinostat	Activation	MCL cell lines	[185]
	Temsirolimus + vorinostat	Activation	MCL cell lines	[185]
NHL, primary effusion lymphoma	CQ	Inhibition	PEL cell lines	[188]
	AG490	Activation	PEL cells	[189]
	Quercetin	Activation	PEL cells	[190]
	BTZ	Activation	PEL cells	[191]
NHL, diffuse large B-cell lymphoma	SIRT3 KO	Activation	DLBCL cells	[194]
	SIRT3	Inhibition	DLBCL cells	[194]
	ATG5 knockdown	Inhibition	DLBCL cells	[194]
	MALAT-1 (lncRNA)	Inhibition	DLBCL cells (Mice)	[195]
	CUL4B	Activation	JNK cells	[197]
NHL, follicular lymphoma	R-CHOP	Activation	FL cells	[199]
	R-bendamustine	Activation	FL cells	[199]
	Bafilomycin A	Inhibition	FL cells	[202]
NHL, T-cell lymphoma	miR-449	Inhibition	T-cell lymphoma	[205]
	Sinensetin	Inhibition	Jurkat cells	[207]
	BCYRN1	Activation	Extranodal NK/T-cell lymphoma cells	[208]
	Crizotinib and CQ	Activation	NHL cells	[210]
	Crizotinib	Activation		[210]
	Metformin	Activation		[211]
	Metformin + doxorubicin or temsirolimus	Activation	NHL cells	[211]
	Fenugreek extract	Activation	Jurkat cells	[212]

Abbreviations: MM, multiple myeloma; CLL, chronic lymphocytic leukemia; ALL, acute lymphocytic leukemia; CML, chronic myeloid leukemia; AML, acute myeloid leukemia; HL, Hodgkin’s lymphoma; NHL, non-Hodgkin’s lymphoma; FLT3-ITD, FMS-type tyrosine kinase 3 gene and internal tandem duplication; MCL, mantle cell lymphoma; PEL, primary effusion lymphoma; DLBCL, diffuse large B-cell lymphoma; FL, follicular lymphoma.

## 6. Germline Variation in Autophagy-Related Genes

As mentioned above, it is widely known that the aberrant expression of autophagy-related genes is associated with cancer development [211,212] and that multiple activators of autophagy or specific autophagy-related genes are commonly found in cancer-associated regions [213,214] (Table 2). Interestingly, it has also been described that autophagy genes are commonly mutated in both solid cancers [215,216] and hematological malignancies [217], and that their regulation influences not only the risk of developing malignant diseases [217], but also response to conventional treatments, disease progression, and patient survival [218]. Even though multiple studies have evaluated the role of rare variants in modulating autophagy, little is known about the impact of germline variation on the regulation of this fundamental process. 

### 6.1. Common Germline Variation Affecting the Risk of Developing Hematological Malignancies

Recent studies have identified multiple pathogenic variants in autophagy-related genes that are involved in determining the risk of developing autoimmune diseases [219,220,221,222], cardiovascular diseases [223,224], neurodegenerative disorders [225,226,227], and different types of cancer (reviewed in [30] and later updated in [228,229,230]). However, despite the above evidence suggesting a key role of autophagy in the etiopathogenesis of hematological malignancies, the existence of a genetic component controlling this catalytic process in hemopathies has been poorly studied and is restricted to specific diseases [231,232,233].

As for most cancers, the importance of autophagy in the biology of hematological diseases has been demonstrated through the development of genome-wide association studies (GWAS) in large patient populations [233]. These studies have transformed our understanding of hematological diseases. In MM, for instance, these studies have suggested a model of susceptibility based on the transcriptional dysregulation of B cells, in which autophagy seems to play a fundamental role. The most recent studies consistently demonstrate that the presence of biologically functional genetic polymorphisms in mTOR-related genes, such as ULK4, ATG5, and WAC, or in genes related to IRF4–MYC-dependent autophagy, such as CDCA7L, DNMT3A, CBX7, and KLF2, is strongly associated with the risk of developing MM [231]. The expression of ATG5 is very high in plasma cells and seems to be essential both for the correct functioning of autophagy and for the survival of B cells. The ATG5 gene is found in the 6q21 region, and this locus also contains a transcriptional repressor, PRDM1, which has also been shown to be crucial to the survival of plasma cells [233]. On the other hand, ULK4, located in the 3p22.1 region, has also been associated with the risk of developing monoclonal gammopathy of undetermined significance (MGUS), a precursor condition that increases the risk of suffering MM [234]. This polymorphism could, therefore, influence MM through the initiation of MGUS. On the other hand, recent studies have suggested that germline variation within the ATG2B/GSKIP region containing the 14q32 duplication predisposes to early clonal hematopoiesis that, thus, could lead to the development of different myeloid neoplasms [235,236].

### 6.2. Common Germline Variation Affecting Disease Progression, Drug Response, and Patient Survival in Hematological Malignancies

In addition to their effect on disease risk, common germline variations have been linked to disease progression, patient survival, and specific clinical complications such as infections and secondary hematological malignancies. Genetic variants within the ATG16L2 locus have been associated with poorer response to therapy in leukemia, whereas genetic polymorphisms within the NOD2 gene have been associated with severe acute graft versus host disease (GVHD) and treatment-related mortality in T-cell-depleted hematopoietic stem-cell transplantation patients [237]. Recent GWAS and candidate gene association studies have also identified noncoding SNPs located within autophagy-related genes (PRDM1 and ATG1 genes on chromosome 6q21) as risk factors for secondary malignancies in patients formerly treated with radiotherapy for pediatric Hodgkin’s disease [238,239]. Although interesting, the abovementioned genetic studies have only started to shed light on the role of autophagy in the modulation of the risk of developing hematological malignancies and disease progression; therefore, there is a long way to go before we can determine how common variation influences the regulation of autophagy and, thus, disease progression and patient survival. 

**Table 2 cancers-14-05072-t002:** Autophagy-related genes in hematological malignancies.

Disease	Autophagy-Related Genes	References
MM	*ATG2B, ATG4, ATG5, ATG7, ATG14, Akt, Beclin-1, Bnip3, CDCA7L, CBX7, DNMT3A, FOXO3a, KLF2, LC3, Linc00515, miR-140-5p, mTORC1, mTORC2, NBK/Bik, p62, PRDM1, STAT3, ULK1, ULK4, Vsp34, WAC*	[86,89,90,91,92,93,96,98,232,233,235,238]
CLL	*AMPK, ATG5, ATG7, Bag-1, Bax, Beclin-1, Bim, CD38, LAMP2, LC3B, mcl-1, P53, p62, SLAMF1, ZAP70*	[102,104,105,109,110,111,112]
ALL	*ATG12, Beclin-1, BCL2A1, Bax, LC3-II, MYC, ULK2*	[120,122,127]
CML	*ATG3, ATG7*	[135,138]
AML	*ATF4, ATG3, ATG4D, ATG5, ATG7, Beclin-1, CXCR4, LC3, p62, ULK1*	[147,148,149,150,155]
HL	*ATG1, ATG5, Beclin-1, DRAM2, LC3, PRDM1, RORC*	[161,162,163,165]
NHL, B-cell lymphoma	*Beclin-1*	[168,169]
NHL, Burkitt’s lymphoma	*BAFF, LC3*	[171,175]
NHL, mantle cell lymphoma	*ATG5, Caspase-3, CD74, LC3*	[181,183,185]
NHL, primary effusion lymphoma	*ATG5, Beclin-1, STAT3*	[189,190,191]
NHL, diffuse large B-cell lymphoma	*ATG5, CUL4B, LC3, MALAT-1, p62, PCYT1A, SIRT3*	[194,195,196,197]
NHL, follicular lymphoma	*ATP6V1B2, BCL-2, LC3A*	[200,201,202]
NHL, T-cell lymphoma	*MiR-449, BCL-2, BCYRN1*	[205,208,210]

## 7. Conclusions

Autophagy plays an important role in the initiation and progression of many hematological neoplasms, influencing and, in turn, being affected by multiple therapeutic agents currently available. Although great efforts have been devoted to understanding the specific role of the autophagic pathway in different hematological tumors, there is still a long way to go to achieve success with combined therapies based on this molecular mechanism. The promising results achieved by the modulation of autophagy in numerous clinical trials may open new alternatives for more efficient treatments. However, the role of autophagy remains ambiguous in many cases, even playing opposite roles, which suggest that its modulation through therapeutic strategies should be exercised with caution. The discovery and testing of new autophagic modulators, including autophagy-related SNPs or mutations, a complete understanding of the involvement of autophagy in cell transformation and response to treatment, and knowledge of its role in healthy and malignant HSCs will undoubtedly help us to better appreciate the therapeutic potential of this central mechanism in hematological tumors.

## Figures and Tables

**Figure 1 cancers-14-05072-f001:**
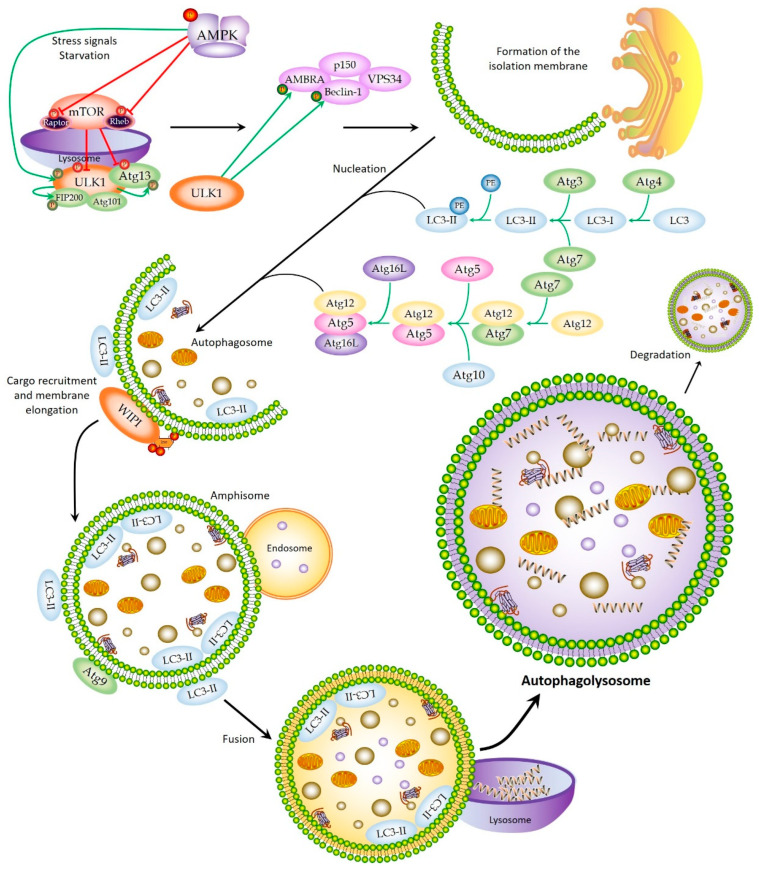
Autophagy pathway. Schematic representation of the autophagy pathway. This complex mechanism is activated by stress signals or starvation and proceeds through a series of steps, including formation of the isolation membrane; membrane nucleation, elongation, and completion of the autophagosome; autophagosome maturation by fusion with a lysosome; and, finally, degradation of autophagic cargoes and recycling of the resulting molecules.

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
