# Peer review of "Autophagy in Hematological Malignancies"

_cancers, 2022, doi:10.3390/cancers14205072_

Round 1
Reviewer 1 Report
In this paper, authors summarize autophagy’s double edge sword role in hematological malignancies, as autophagy can promote survival or induce cytotoxicity in cancer cell. This summary will help researchers target autophagy process for the therapy of blood cancer in the future. Moreover, this review is well-written and authors are knowledgeable in the field of autophagy and cancer research. However, before publication, authors must address minor questions below.
1. Line 149, “causing its activation” should be replaced by “activation of ULK1”, because mTOCR1 is the subject of this sentence.
2. Table 1, for each example authors listed in this table, authors must cite corresponding reference.
3. Line 668, “Autophagy is associated…” should be replaced by “Up-regulated autophagy is associated”.
4. Line 673, I checked corresponding reference here, It should be “Beclin-1-deficient mice” but not “myeloid-deficient mice”. Authors should reorganize this sentence here.
Author Response
Authors' Responses to Reviewer's Comments (Reviewer 2)
In this paper, authors summarize autophagy’s double edge sword role in hematological malignancies, as autophagy can promote survival or induce cytotoxicity in cancer cell. This summary will help researchers target autophagy process for the therapy of blood cancer in the future. Moreover, this review is well-written and authors are knowledgeable in the field of autophagy and cancer research. However, before publication, authors must address minor questions below.
Comment #1: Line 149, “causing its activation” should be replaced by “activation of ULK1”, because mTOCR1 is the subject of this sentence.
Reply: We thank the reviewer for this comment. We have amended that sentence. In addition, we have extensively revised English grammar through a professional English editing service.
Comment #2: Table 1, for each example authors listed in this table, authors must cite corresponding reference.
Reply: As suggested by the reviewer, we have added references to table 1. We agree with the reviewer that the addition of references to the table 1 will facilitate the reading of the manuscript.
Comment #3: Line 668, “Autophagy is associated…” should be replaced by “Up-regulated autophagy is associated”.
Reply: We thank the reviewer for this comment. We have amended that sentence.
Comment #4: Line 673, I checked corresponding reference here, it should be “Beclin-1-deficient mice” but not “myeloid-deficient mice”. Authors should reorganize this sentence here.
Reply: We thank the reviewer for this comment. We have also amended that sentence.
Reviewer 2 Report
In this review article, Olga García Ruiz and collogues wrote a very informative review on the importance of autophagy in hematological malignancies. They nicely introduced autophagy-related mechanisms and subsequently their roles in different hematological malignancies as well as in normal hematopoiesis. To improve the flow of the reading some editing is required.
The comments are as follows:
Comments:
1. There are multiple spelling errors that should be corrected. For example, in Figure 1, “Lysosome” spelling is wrong. In line 104; should be “dysregulated”. In line 355; should be “Bnip3”. A careful revision is required.
2. A separate table with different autophagy-related genes and the hematological disease they are associated with will be helpful. The same gene, like Beclin-1, can be associated with many hematological diseases as the authors pointed out. A table with proper references to the genes and the associated diseases will help the readers. The authors should include the references in the table.
3. For Table 1: “Description of autophagy findings in different hematological malignancies.”; the authors should include the references in the table.
4. A brief figure legend for Figure 1 will be helpful.
5. The introduction should be more precise. The authors wrote the autophagy process in the introduction (lines 72-80) which can go to section 2 and that will be easy to follow. Similarly, in section 2 lines 128-134 are not needed as they are described in the following sections. Good editing is required to delete the repetitive information which will maintain the flow of the reading.
6. The entire conclusion section is in bold. Font correction is needed. Also in lines 970, and 971 the authors wrote “main” which should be “many”.
Author Response
- There are multiple spelling errors that should be corrected. For example, in Figure 1, “Lysosome” spelling is wrong. In line 104; should be “dysregulated”. In line 355; should be “Bnip3”. A careful revision is required.
Reply: We thank the reviewer for the careful revision. We have corrected all spelling errors in both figures and text.
- A separate table with different autophagy-related genes and the hematological disease they are associated with will be helpful. The same gene, like Beclin-1, can be associated with manyhematological diseases as the authors pointed out. A table with proper references to the genes and the associated diseases will help the readers. The authors should include the references in the table.
Reply: According to the reviewer suggestion we have added a table including autophagy-related genes by disease. The table has been numbered as Table 2 and it includes references.
- For Table 1:“Description of autophagy findings in different hematological malignancies.”;the authors should include the references in the table.
Reply: As suggested by the reviewer, we have added references to table 1.
- A brief figure legend for Figure 1 will be helpful.
Reply: As suggested by the reviewer, we have added a brief legend to the Figure 1. It summarizes what it is described in the figure.
- The introduction should be more precise. The authors wrote the autophagy process in the introduction (lines 72-80) which can go to section 2 and that will be easy to follow. Similarly, in section 2 lines 128-134 are not needed as they are described in the following sections. Good editing is required to delete the repetitive information which will maintain the flow of the reading.
Reply: We agree with the reviewer. We have moved that specific paragraph from section 1 to section 2. We have also deleted the repetitive information from section 2 (former lines 128-134).
- The entire conclusion section is inbold.Font correction is needed. Also in lines 970, and 971 the authors wrote “main” which should be “many”.
Reply: We have amended the format of the conclusion section and we have corrected spelling errors.